# A Case-Based Approach to New Directions in Dietary Therapy of Crohn’s Disease: Food for Thought

**DOI:** 10.3390/nu12030880

**Published:** 2020-03-24

**Authors:** Arie Levine, Wael El-Matary, Johan Van Limbergen

**Affiliations:** 1Wolfson Medical Centre, Sackler Faculty of Medicine, Tel Aviv University, Holon 5822012, Tel Aviv, Israel; 2Section of Pediatric Gastroenterology & Nutrition, University of Manitoba, Winnipeg, MB R3E 0Z2, Canada; welmatary@hsc.mb.ca; 3Amsterdam University Medical Centres, Emma Children’s Hospital, 1105 AZ Amsterdam, The Netherlands; j.e.vanlimbergen@amsterdamumc.nl; 4Tytgat Institute for Liver and Intestinal Research, Amsterdam Gastroenterology and Metabolism, Academic Medical Centre, University of Amsterdam, 1105 AZ Amsterdam, The Netherlands

**Keywords:** Crohn’s disease, microbiome, treatment, inflammatory bowel disease, diet, Crohn’s disease exclusion diet (CDED)

## Abstract

Recent evidence has demonstrated that Crohn’s disease may have its roots in dysbiosis of the microbiome and other environmental factors. One of the strongest risk factors linked to immune activation appears to be diet. Exclusion diets have been shown to ameliorate inflammation and induce remission in 70–80% of treatment-naïve children at disease onset, and to induce remission in patients that lose response or are refractory to currently recommended medical therapy. Recent studies have also linked dietary modulation of the microbiome with clinical remission, while reintroduction of the previous habitual diet led to reactivation of inflammation and reversion of the dysbiotic state. While dietary therapy has usually been used as a first line therapy as a bridge to immunomodulators, newer insights suggest that new treatment paradigms involving dietary therapy may allow different treatment strategies. This case-based narrative review will discuss the Crohn’s disease exclusion diet (CDED) as monotherapy, combination therapy with drugs, as a rescue therapy in refractory patients and for de-escalation from medical therapy.

## 1. Introduction

The incidence and prevalence of the chronic inflammatory bowel diseases Crohn’s disease (CD) and ulcerative colitis (UC) have risen significantly in the last two decades. Over 3.5 million people in the US and Europe suffer from one of these two diseases, which are at present chronic and incurable [1].

Recent data suggest that the inflammation in Crohn’s disease may be driven by dietary factors that may negatively alter the microbiome, gut barrier or innate immunity [2,3,4]. This concept is supported by epidemiological studies linking Western diets with disease, and conversely demonstrating reduction of disease incidence among patients consuming a more Mediterranean diet [2,5]. This has led to a resurgence of interest in dietary therapy. Until recently, evidence-based dietary therapy in CD was limited to exclusive enteral nutrition (EEN), which appears to act by exclusion of unidentified dietary components that affect the gut microbiome or gut barrier, and was found to be superior to corticosteroids for induction of remission and improvement of mucosal inflammation [6,7,8]. EEN is a highly effective induction of remission therapy, capable of inducing remission in 60–80% of patients with new onset luminal CD. It is, however, difficult to perform, requires significant effort by the staff to keep children on exclusive formula without table food, and has no effective maintenance strategy. If diet is triggering inflammation, a more sustainable and patient-friendly dietary approach is required that would allow long-term reduction in exposure and allow sustained remission and mucosal healing. A long-term strategy would have the potential for dietary monotherapy to reduce the burden of immune suppression, and a patient friendly diet would allow more patients to use dietary therapy.

The next leap, towards more effective and longer-term dietary therapy, was identification of the plausible dietary culprits and development of a dietary therapy based on exclusion of these components while allowing access to whole foods that would be safe. The Crohn’s disease exclusion diet (CDED), with or without supplemental partial enteral nutrition (PEN), is a next-generation dietary therapy that was shown to be as effective as EEN for induction of remission and reduction in inflammation, but with better tolerability and sustained remission in a randomized controlled trial [3]. It also identified how dietary therapy might affect the microbiome by altering dysbiosis, act to decrease intestinal permeability, and demonstrated that predesigned dietary manipulation of the microbiome to achieve a clinical goal can be a successful strategy. This recent study challenges existing concepts of optimal therapy in CD and raises the question of how dietary therapy with EEN and the CDED should be employed and might be employed in the future. The following case-based discussion reviews some of these concepts.

## 2. Principles of the Crohn’s Disease Exclusion Diet 

### Dr. Wael El-Matary: Dr Levine, how would you Describe the Crohn’s Disease Exclusion Diet in a Nutshell?

Dr. Arie Levine: Exclusive enteral nutrition, consisting of a liquid polymeric formula without access to other food for 6–8 weeks has been the mainstay of dietary therapy for more than a decade. It is useful for induction of remission but is not sustainable. The CDED is a high-protein low-fat multistage diet designed to exclude dietary components that generate a high-Proteobacteria/low-Firmicutes-led dysbiosis, components that may allow biofilms or mucosal bacteria, as well as components that impair barrier function or bacterial clearance mechanisms [2]. It also provides mandatory sources of beneficial fiber. It was designed with a 12-week induction phase and third stage maintenance phase that allows gradual access to more foods every six weeks and free meals. This diet involves ordinary whole unprocessed foods such as fruits and vegetables, chicken, eggs, rice and potatoes and can be coupled with liquid formulas to provide additional sources of protein and calcium to ensure growth and restitution of lean body mass [2,3]. We recently demonstrated that it was as effective as EEN for induction of remission, and was superior to an EEN-based strategy for sustained remission and reduction in inflammation. At a microbiological level, it degraded Proteobacteria and increased Firmicutes and Bacteroides while restoring intestinal permeability [3]. The first phase is given for six weeks, it is lower in fiber than phase 2 as luminal narrowing may be present until inflammation starts to subside. The first stage does not include and foods that we perceived to have a potential deleterious effect. The second phase is a step-down phase that includes more fruits and vegetables, such that by week 10 almost all fruits and vegetables are allowed. It also introduces limited specified amounts of bread and red meat (potentially deleterious foods) and legumes (have the potential to aggravate symptoms) to improve quality of life.

## 3. How Effective is the Diet?

### 3.1. Dr. Wael El-Matary, Editor

The recently published paper in Gastroenterology [3] demonstrated that CDED with PEN is very effective for induction of remission and reduction of inflammation for 12-weeks in children with mild to moderate new onset Crohn’s disease. Dr. Levine, how effective is it for longer-term goals, or in patients with severe disease? Can children and adults maintain the diet over time?

#### 3.1.1. Case 1

The first case was a previously healthy 18-year-old male presented with a three-month history of abdominal pain, vomiting, weight loss and intermittent diarrhea. His initial C reactive protein (CRP) was 110 mg/L (normal less than 5 mg/L) and fecal calprotectin was 5000 µg/g (normal less than 100 µg/g), with a normal hemoglobin albumin and transaminases. A gastroscopy demonstrated aphthous ulcers in the duodenal bulb but was otherwise normal. A colonoscopy revealed aphthous ulcers along the colon with linear ulcers in the terminal ileum, biopsies demonstrated chronic inflammation and epithelioid granulomas. A magnetic resonance enterography (MRE) identified enhancement of a short segment of the terminal ileum without stricture or fistula. Infliximab was recommended by their physician but the family refused and approached me for dietary therapy. As the patient did not have complicated disease, we agreed that we would start with a six-week trial diet and decide if he needed additional medications based on his response. The patient started the CDED with PEN. After six weeks of therapy he was in clinical remission, his hemoglobin was 16.6 g/dL, his CRP was normal (< 5 mg/L). A calprotectin at week 12 was 8 µg/g. The patient transitioned to the phase 3 maintenance diet without drugs and remained in remission during the year. Fifteen months after starting the diet, he had a colonoscopy performed that demonstrated complete mucosal healing, with normal biopsies of the colon, an ileal biopsy demonstrated mild focal active inflammation.

#### 3.1.2. Case 2

The second case was a 10-year-old girl who presented with abdominal pain and a weight loss of 2 kg over three months. Laboratory tests performed demonstrated persistently elevated CRP (22 and 24 mg/L), a fecal calprotectin of >300 µg/g. An abdominal ultrasound noted thickening of the terminal ileum with enlarged lymph nodes. A gastroscopy demonstrated thickened gastric folds with evidence for chronic inflammation in biopsies, but no ulcers; *Helicobacter pylori* was not present. A colonoscopy revealed aphthous ulcers in the transverse colon with linear and aphthous ulcers in the ascending colon, while the terminal ileum was reported as normal. Biopsies demonstrated chronic inflammation consistent with Crohn’s disease. An MRE demonstrated thickening and enhancement of the bowel wall in the distal ileum. Her pediatric Crohn’s disease activity index (PCDAI, remission defined as <10) upon presentation to our clinic was 30, consistent with moderately active disease. She refused medical therapy and was started on the CDED with PEN. After six weeks she was in clinical remission with normal CRP and had regained 1.3 kg. She performed the second phase of the induction diet and then the phase 3 maintenance diet. During the subsequent 12 months remained in clinical remission with normal CRP and fecal calprotectin. An MRE and colonoscopy were repeated between 12–15 months, both were completely normal at this time. She remained in sustained deep remission for three years, maintaining the diet with some difficulty as she struggled with adherence at times. During the ensuing summer she travelled abroad several times and did not adhere to the diet. Though she felt well her calprotectin increased from 16 to 300 µg/g. She regressed to the phase 1 induction diet for four weeks and then returned to the maintenance diet. Her calprotectin normalized and a subsequent ileocolonoscopy was completely normal.

Dr. Arie Levine: These two cases serve as examples of how dietary therapy with the CDED and PEN might be used in the future and impact disease. In patients with mild to moderate uncomplicated disease, CDED can be used as monotherapy for remission and maintenance of remission. In the clinical trial, roughly 80% of patients entered clinical remission with diet by week 6, and normal CRP remission and sustained remission were present in 75% of patients at week 12. In both these new onset cases, no drugs were required and complete mucosal healing was obtained just with diet. Ultimately, long-term dietary monotherapy requires a motivated patient, in less motivated patients, drug and diet are usually combined. In the second case, when the diet was stopped after three years, rebound inflammation ensued without clinical relapse, and regressing to the first stage for four to six weeks was effective to re-induce remission or reduction in inflammation, and the maintenance diet could be continued. Maintenance of the diet or transition to a healthy lifestyle over time is feasible in motivated patients, but even those patients will go on vacations and temporarily stop the diet, which is very reasonable in our practice. We discuss this with the patients and allow them to enjoy their vacations, but if they have resulting symptoms or an increase in inflammation, we use the first phase diet to regain remission and then go straight to maintenance. The first case also demonstrates that diet can work very well even with severe inflammation. The efficacy of the diet in isolated colonic disease has not been addressed in our studies thus far. It is unclear at present if the changes in the microbiome of colonic only disease (particularly left sided colonic disease) are similar to those observed with ileocolonic and ileal disease. Our experience shows that patients with isolated colonic disease also respond to the diet, though my impression is that the remission rate is lower. We clearly need better microbiological data based on site of the disease and better data for the CDED in isolated colonic left-sided disease.

### 3.2. Dr. Wael El-Matary, Editor

Dr Van Limbergen, could you provide more insight as to which patients would be best suited to use or avoid dietary therapy with the CDED and when you should combine diet with drugs as opposed to monotherapy?

#### Case 3

A 15-year-old male presented with acute-on-chronic right lower abdominal pain, diarrhea, rectal bleeding, fever and weight loss with delay in seeking medical attention until this acute presentation. A right lower quadrant inflammatory mass was palpable. Inflammatory markers were elevated (CRP > 300) with evidence for iron-deficiency anemia. Imaging of the abdomen showed two abscesses in the right lower quadrant with an entero-enteral fistula, of which the more superficial one, was amenable to percutaneous drainage by means of a pig-tail drain. Endoscopic investigations showed ulceration and stenosis in the terminal ileum. MRI Pelvis showed an inter-sphincteric fistula complex. We would typically treat penetrating intestinal disease with early surgical intervention after a period of antibiotic therapy. In view of the complex perianal disease, we decided not to perform intestinal surgery at that stage: EEN and intravenous antibiotics (Amoxicillin/clavulonic acid/Gentamicin) were started and anti-Tumour Necrosis Factor (TNF) therapy (infliximab) was added after drainage and improvement. Echographic monitoring of abscesses showed marked improvement and the drain was removed after a few weeks. CDED+PEN replaced EEN after eight weeks. After four months, MRE showed resolution of penetrating ileal disease without evidence of a fistula (with evidence of a short segment of thickening and mild prestenotic dilatation but no hypervascularity). At the latest clinical follow-up, perianal disease was quiescent without a visible external fistula or drainage.

Dr Johan Van Limbergen: This third case illustrates a few key advantages of dietary therapy. In this case, due to intestinal penetrating disease, we used EEN for induction and only moved to CDED+PEN for maintenance, as immune suppression could not be started immediately due to a septic complication. Effective induction with dietary therapy can act as a bridge to starting immune suppression and allows for a period of anti-infectious therapy, optimization of vaccination status (which are required before starting immune suppression).

Using diet in combination with judiciously timed immune suppression facilitates nutritional rehabilitation with a maintenance strategy of long-term dietary modification in mind. Nutritional therapy has been shown to help maintain response to immune suppression [9,10,11,12,13]. In cases of penetrating ileal disease, a period of EEN in combination with antibiotic therapy, can lead to mucosal healing, reduce extent of inflammation, and help avoid parenteral nutrition (and its associated risks). In cases of stricturing disease, the degree of intestinal inflammation and stenosis can make EEN (and use of nasogastric tube feeds if needed) the better option. As in this case, it is key to start adequate immune suppression once infectious complications are addressed. The multidisciplinary IBD team can then supervise a gradual transition from EEN towards more solid food in CDED+PEN.

Many of our patients present with pubertal delay and concerns regarding (skeletal) growth. Avoidance of prolonged steroid exposure reduces health/psychological concerns with regard to body composition, skeletal and mental health. Since EEN has been proven to be as effective or better than steroids, at least in children, dietary therapy holds promise as a steroid-sparing agent [6,8,14].

In all stages of therapy, it is important to avoid conflict when choosing the optimal combination of nutritional and medical therapy. The CDED + PEN then provides nutritional maintenance to help sustain the effect of immune suppression. In this setting, we often personalize dietary advice to make CDED+PEN more or less strict depending on imaging, calprotectin, inflammatory markers and stage of disease. In order to improve adherence, a multidisciplinary team approach is most effective. Clinical care pathways can help optimize communication with patients and families as well as multidisciplinary decision-making. Online resources of nutrients and recipes for each phase have been developed and can help facilitate communication between caregivers and patients (mymodulife.com, modulifexpert.com).

It then remains important to position diet as a sustainable lifestyle intervention, used in combination with drugs when needed—not simply to avoid drugs. In order to improve adherence, it may be helpful to show that diet is a controllable variable by the young patient, focusing e.g., on parallels with athletes’ dietary choices and nutritional and lifestyle recommendations for cardiovascular/metabolic/cancer-risk reduction.

Dr Arie Levine: I would add that there are limitations to the success of dietary therapy. It is less effective for extraintestinal manifestations and very likely in smokers. Smoking drives dysbiosis-decreasing Firmicutes so patients who wish to use dietary therapy must stop smoking during the induction phase [15]. Compliance is a major issue and a specific dietary program with a dietitian and a support system needs to be in place and to encourage patients to try diet.

### 3.3. Dr Wael El-Matary, Editor

You have shared cases with us in which dietary therapies such as EEN and CDED were used as monotherapies or as a bridge to immune suppression. How else has dietary therapy changed your treatment paradigm?

Dr. Johan Van Limbergen: Dietary therapy helps address issues that are very relevant and prevalent with existing therapies targeting immune cells and pathways. It addresses the microbiological issues triggering inflammation as opposed to suppressing inflammation, so it is very safe. It addresses the nutritional deficiencies. It addresses safety issues. Its safety and efficacy suggest that future combination therapy could involve a drug and diet instead of two immune suppressing drugs. The inflammatory microbiome, gut barrier defects and inflammation would be better addressed, without using two immune-suppressive medications. Combination therapy with anti-TNF and an immune modulator increases the risk of cancer. Reducing the burden of these drugs in children and adults should be an important goal. Diet helps address these issues. Studies in adults have shown the benefit of combining anti-TNF therapy with a dietary strategy to prolong remission [10,16].

Dr Arie Levine: I totally agree. In addition, we believe that dietary therapy allows new treatment strategies. We are successfully using dietary therapy with the CDED as a rescue induction therapy for biologic refractory patients, and as a strategy for de-escalation from drugs. We have published a case series using CDED as a successful induction therapy for patients with refractory disease or loss of response to biologics [9]. If diet is indeed the trigger for inflammation in many patients, it might be possible to de-escalate them from therapy if they maintain a diet or lifestyle. We often use this to actually convince patients to start an immune-modulator with the diet. We will stipulate that if there is complete mucosal healing after one year of drug and diet, and the patient is willing to continue the maintenance diet, we would be willing to stop the immune-modulator and see if we can maintain mucosal healing. If this is successful, we maintain diet alone, but if it is unsuccessful, we will recommend moving to a biological medication. However, the response of multidrug refractory disease is clearly the most gratifying new niche for dietary therapy as the following case discloses.

#### Case 4

A 16-year-old patient of mine had multidrug refractory disease but had always refused to diet. He was diagnosed at the age of 13 with panenteric and perianal disease. He was started on combination therapy with antibiotics infliximab and azathioprine for six months after which he stopped azathioprine; unfortunately, he lost response shortly after. He was then transitioned to adalimumab monotherapy followed by weekly adalimumab with methotrexate. He was a primary nonresponder to this combination despite high trough levels. His disease became more active and he was therefore switched to ustekinumab. He was a primary nonresponder to ustekinumab as well after six months, in a continuous drug resistant flare. We discussed surgery as the next option. Fate intervened and he was hospitalized with a severe flare, abdominal pain, frequent diarrhea, significant weight loss, hypoalbuminemia, CRP of 120 g/L and fecal calprotectin of 1300. He received two weeks of exclusive enteral nutrition followed by the CDED with PEN 50% for the next 10 weeks. By the third week he was asymptomatic for the first time in eight months. At week 6, his CRP had declined from 120 to 29 g/L, his albumin had increased from 33 to 42 g/L, and his PCDAI had declined from 47.5 to 5. His calprotectin was repeated at week 14 and it had declined from 1300 to 263 µg/g. He is continuing on ustekinumab 90 mg every eight weeks with the maintenance diet and, at this time, five months have elapsed and he is still in remission.

## 4. Conclusions

Dr Arie Levine and Dr. Johan van Limbergen: Our increasing understanding of the role of diet in instigating inflammation is already opening up new treatment strategies (Figure 1). Diet may be used as monotherapy, as combination therapy, for de-escalation of drugs and as a rescue therapy for refractory patients. The clear advantage is reduction in exposure to drugs while addressing the source of inflammation. The advances in microbiome sequencing and analysis have shown clearly that dysbiosis is a complex change of the microbiome community in which Proteobacteria are associated with disease, primary nonresponse to diet as well as recurrence of mucosal inflammation [3]. In the future, a better characterization of the inflammation-associated dysbiosis will help to characterize which patients might benefit more from either increased immune suppression, antibiotics, dietary intervention or combinations of these strategies to achieve mucosal healing.

We have discussed clinical goals of dietary therapy and highlighted that dietary therapy may be more potent than was previously appreciated. We both believe that we should expand our horizons beyond current clinical goals with next generation microbial modulating therapies and do a better job of addressing the source of inflammation. Though this remains a hypothetical until more studies are performed, perhaps we should be trying to achieve correction of dysbiosis as a microbiological goal. Dietary therapies such as the CDED show promise in this hitherto unexplored field.

## Figures and Tables

**Figure 1 nutrients-12-00880-f001:**
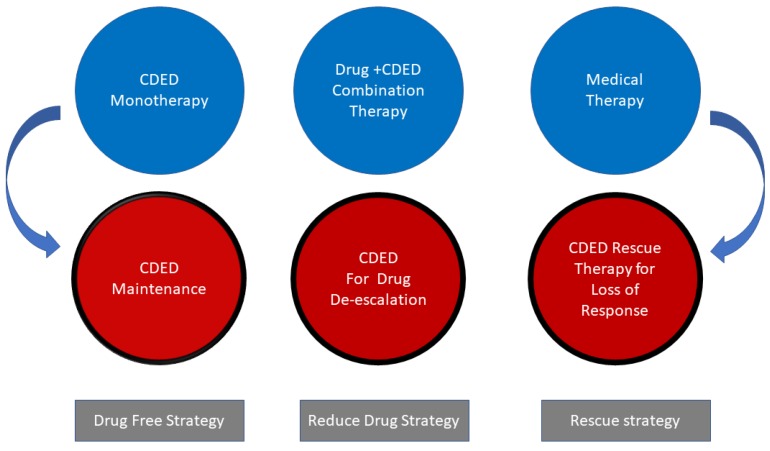
New potential strategies incorporating dietary therapy in Crohn’s disease. CDED: Crohn’s disease exclusion diet.

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
