# Peer review of "A Case-Based Approach to New Directions in Dietary Therapy of Crohn’s Disease: Food for Thought"

_nutrients, 2020, doi:10.3390/nu12030880_

Round 1

Reviewer 1 Report

The manuscript entitled “A Case based Approach to New Directions in Dietary Therapy of Crohn’s Disease: Food for thought” by Arie Levine et al. addresses a very interesting therapeutic approach in Crohn disease. The recently published studies are very promising and therefore this present article that includes patient cases is very helpful for the readership of Nutrients. 

Taken together the here presented manuscript is well written and strengthen our evidence that special dietaries could significantly help to improve the treatment of Crohn disease.

Author Response

Thank you

Reviewer 2 Report

The authors are experts in dietary therapy for Crohn’s disease and provide a case-based discussion of 4 patients treated with the revolutionary Crohn’s disease elimination diet and provide insight into important components of the diet related to each of the cases. This is a very useful paper for practicing clinicians as the potential advantages of dietary therapy are illustrated and this can be difficult to appreciate for those who are not experienced in its use. I have the following suggestions:

1. The format of using the cases to illustrate important points is useful although the use of the question asked by the editor followed by the case and then the expert answers can be a bit difficult to follow in written form. Perhaps headings for each section in relation to the problems addressed by the case and then presenting the cases followed by the questions and answers may improve the readability. For example, for cases 1 and 2, the heading of “Long-term use of dietary therapy” may assist readers in understanding what is addressed by the cases.

2. For cases 1 and 2, both patients had colonic disease endoscopically. Can the authors provide some comment of the efficacy of the CDED in large bowel disease versus small bowel disease?

3. In the nutshell summary of the diet some mention of the different phases of the diet would be of value to understanding the subsequent discussions.

4. A table of key messages or concepts would be of value or an image of where the authors perceive dietary therapy has a role in the treatment algorithm would help readers.

Minor points:
Line 147 – use generic names for antibiotics

Line 151 – consider changing “evidence for a fistula” to “evidence of a fistula”

Line 200 – “Anti” should be lower case

Line 210/212 – consider changing “immune-modulator” to “immunomodulator”

Author Response

Thank you for your their kind comments and for affording us the opportunity to improve this manuscript.

Our replies are below .

Reviewer 2:

1.We have added headings before the questions as suggested

 2.The first two cases were actually ileocolonic but we have addressed the reviewers thoughts on colonic disease. The following sentences were added to discuss the issue of isolated colonic disease:

The efficacy of the diet in isolated colonic disease has not been addressed in our studies thus far. It is unclear at present if the changes in the microbiome of colonic only disease (particularly left sided colonic disease) are similar to those observed with ileocolonic and ileal disease. Our experience shows that patients with isolated colonic disease also respond to the diet, though my impression is that the remission rate is lower. We clearly need better microbiological data based on site of the disease and better data for the CDED in isolated colonic left sided disease.

 3.We have added the following sentences to the diet in a nutshell paragraph:

The first phase is given for 6 weeks, it is lower in fiber than phase 2 as luminal narrowing may be present until inflammation starts to subside. The first stage does not include and foods that we perceived to have a potential deleterious effect. The second phase is a step- down phase that includes more fruits and vegetables, such that by week 10 almost all fruits and vegetables are allowed. It also introduces limited specified amounts of bread and red meat (potentially deleterious foods) and legumes (have the potential to aggravate symptoms) to improve quality of life.

4.We have added a figure about new strategies and added the following section to the conclusions: Our increasing understanding of the role of diet in instigating inflammation is already opening up new treatment strategies (figure 1). Diet may be used as monotherapy, as combination therapy, for de-escalation of drugs and as a rescue therapy for refractory patients. The clear advantage is reduction in exposure to drugs while addressing the source of inflammation. The advances in microbiome sequencing and analysis have shown clearly that dysbiosis is a complex change of the microbiome community in which Proteobacteria are associated with disease, primary non-response to diet as well as recurrence of mucosal inflammation.

 5.We have corrected these

Thank you once again

Sincerely

Arie Levine – Corresponding author